# Per- and Poly-Fluoroalkyl Substances in Portuguese Rivers: Spatial-Temporal Monitoring

**DOI:** 10.3390/molecules28031209

**Published:** 2023-01-26

**Authors:** Marta O. Barbosa, Nuno Ratola, Vera Homem, M. Fernando R. Pereira, Adrián M. T. Silva, Ana R. L. Ribeiro, Marta Llorca, Marinella Farré

**Affiliations:** 1LSRE-LCM—Laboratory of Separation and Reaction Engineering-Laboratory of Catalysis and Materials, Faculty of Engineering, University of Porto, Rua Dr. Roberto Frias, 4200-465 Porto, Portugal; 2ALiCE—Associate Laboratory in Chemical Engineering, Faculty of Engineering, University of Porto, Rua Dr. Roberto Frias, 4200-465 Porto, Portugal; 3Centre for Research and Intervention in Education (CIIE), Faculdade de Psicologia e de Ciências da Educação, Universidade do Porto, Rua Alfredo Allen s/n, 4200-135 Porto, Portugal; 4LEPABE—Laboratory for Process Engineering, Environment, Biotechnology and Energy, Faculty of Engineering, University of Porto, Rua Dr. Roberto Frias, 4200-465 Porto, Portugal; 5ON-HEALTH Research Group, Institute of Environmental Assessment and Water Research (IDAEA-CSIC), C/Jordi Girona, 18-26, 08034 Barcelona, Spain

**Keywords:** PFASs, surface water, risk assessment, liquid chromatography, mass spectrometry

## Abstract

Eighteen per-and polyfluoroalkyl substances (PFASs) were investigated in surface waters of four river basins in Portugal (Ave, Leça, Antuã, and Cértima) during the dry and wet seasons. All sampling sites showed contamination in at least one of the seasons. In the dry season, perfluorooctanoate acid (PFOA) and perfluoro-octane sulfonate (PFOS), were the most frequent PFASs, while during the wet season these were PFOA and perfluobutane-sulfonic acid (PFBS). Compounds detected at higher concentrations were PFOS (22.6 ng L^−1^) and perfluoro-butanoic acid (PFBA) (22.6 ng L^−1^) in the dry and wet seasons, respectively. Moreover, the prospective environmental risks of PFASs, detected at higher concentrations, were evaluated based on the Risk Quotient (RQ) classification, which comprises acute and chronic toxicity. The results show that the RQ values of eight out of the nine PFASs were below 0.01, indicating low risk to organisms at different trophic levels in the four rivers in both seasons, wet and dry. Nevertheless, in the specific case of perfluoro-tetradecanoic acid (PFTeA), the RQ values calculated exceeded 1 for fish (96 h) and daphnids (48 h), indicating a high risk for these organisms. Furthermore, the RQ values were higher than 0.1, indicating a medium risk for fish, daphnids and green algae (96 h).

## 1. Introduction

The presence of per- and poly-fluoroalkyl substances (PFASs) in the environment is a worldwide concern. PFASs comprise a vast class of chemicals that usually have an aliphatic carbon composition, in which hydrogen molecules have been completely or partially (i.e., prefix per- or poly-, respectively) substituted by fluorine. They are characterized by their exceptionally high thermal and chemical stability, thanks to the strength of the carbon-fluorine bond [1]. In addition, PFASs are excellent surfactants, surface protectors and water and oil repellents. Therefore, PFASs have been extensively used since the 1940s in a wide range of consumer and industrial products, including firefighting foams, alkaline cleaners, non-stick cookware, lubricants, adhesives, paints, additives, coatings, among many others [2,3]. However, the concern about the adverse effects of PFASs, such as perfluoro-octane sulfonate (PFOS) and perfluorooctanoic acid (PFOA) did not start until the 1990s [4,5]. These two compounds are the more recalcitrant and have been related to cancer promotion, liver damage and hormone disruption. PFOS and PFOA and their salts, which were phased out by the major producers, are listed under the Stockholm Convention on Persistent Organic Pollutants (POPs). The case of PFOS and perfluoro-octane-sulfonyl fluoride (POSF) are listed under Annex B [6], whereas PFOA and its salts are listed under Annex A [7].

However, as aforementioned, PFASs are a large class of synthetic compounds and different replacement compounds are widely distributed in the market. PFASs, in general, tend to bioaccumulate and bio-magnify through food chains [8,9,10,11] and they have been detected in all environmental compartments (e.g., sediments, surface, drinking water and groundwater) [5,12,13,14,15,16,17,18,19,20,21,22,23]. They are considered “forever chemicals”, due to their persistence [1,3,24]. Given the potential risks associated with PFASs, some actions are currently in place to prevent and minimize environmental and human exposure. In 2016, the United States Environmental Protection Agency (USEPA) published national drinking water guidelines that consider the safe concentration of any combination of PFASs as 70 ng L^−1^ [25,26]. Even more recently, on 18 October 2021, USEPA announced the *Comprehensive National Strategy to Confront PFAS Pollution*, which is a roadmap that is focused on three main directives: research, restrict and remediate [27]. At the European Union (EU) level, the new EU Drinking Water Directive recommends a limiting value of 0.5 µg L^−1^ for total PFAS [28], which is a threshold that is well above that which is recognized in the US. The technical guidelines concerning analysis methods for monitoring of PFASs in the EU are not expected to be defined until January 2024, including frequency of sampling, detection limits and parametric values. Regarding surface water (SW), PFOS and its by-products are included as a priority hazardous substance (Directive 2013/39/EU), with an Environmental Quality Standard limiting value of 0.13 ng L^−1^ in seawater and 0.65 ng L^−1^ in SW [29]. A recent proposal amending this Directive and Directive 2006/118/EC on the protection of groundwater included the sum of 24 PFASs expressed as PFOA-equivalents based on the potencies of the substances relative to that of PFOA, specifically a groundwater Quality Standard of 0.0044 µg L^−1^ as the sum of PFOA equivalents, the same value as an annual average value of 0.0044 µg L^−1^ in inland SW and other SW, and an Environmental Quality Standard for biota and sediments corresponding to 0.077 μg kg^−1^ wet weight. Moreover, two groups of PFASs were identified in 2020 as substances of very high concern (SVHC) under the Registration, Evaluation, Authorisation and Restriction of Chemicals (REACH) regulation and several other PFASs are on the candidate list of SVHC [30]. Nevertheless, it should be highlighted that nowadays, in the EU, only six compounds and their salts are under regulation.

The occurrence of PFASs in SW has been reported across Europe [20,31,32,33,34,35,36,37,38,39,40,41,42,43,44,45,46,47,48,49], but despite this effort in Portugal only two monitoring studies have been carried out by our group [50,51], with PFOS as the target of the PFASs group. However, studies of a wide range of PFASs, including emerging examples, still need to be performed. 

Therefore, the main goal of the present study was to perform a two-seasonal monitoring campaign, during the dry and wet seasons, of 18 PFASs in four Portuguese rivers, namely Ave, Leça, Antuã and Cértima, which are identified as polluted water courses. Moreover, the ecological risk assessment of PFASs that were found at higher concentrations was calculated. To the best of our knowledge, this is the first study characterizing contamination with PFASs and their associated risk in SW of Portugal.

## 2. Results and Discussion

### 2.1. Seasonal Variation and Distribution of PFASs 

PFBA, PFHpA, PFOA, PFNA, PFDA, PFTeA, PFHxS, PFOS and PFBS were detected in this study and PFBA, PFOA, PFNA, PFDA, PFOS and PFBS were found in all four investigated rivers in Portugal in at least one season (Appendix A). Long-chain PFASs, such as PFUnDA, PFDoDA, PFTrDA and PFTeA were less frequent because, nowadays, short-chain PFASs are used as replacements for PFOS and PFOA and their water solubility is higher [52]. On the other hand, the recalcitrant C_8_ compounds, such as PFOS and PFOA, were extensively used for over three decades. This extensive use, together with their high persistence, explains the fact that, even after their discontinuation in production, these compounds continue to be frequently detected. In general, the PFAS concentrations were lower in the winter, as expected, due to the higher flow rates in this season that promote a dilution effect in the river courses. Despite this, for two compounds (PFBA and PFHpA), a contrary tendency was shown. Both chemicals were predominantly detected in the wet season with frequencies of PFBS (76.5%) and PFBA (32.4%), in contrast to 13.8% and 23.5% in the dry season, respectively. Moreover, PFBA has reported the highest concentration in the wet season, with 22.6 ng L^−1^ in one of the sites of the Antuã River. This fact can be attributed to the rain wash from surrounding soils and the higher water solubility of short-chain compounds.

#### 2.1.1. Ave River 

The Ave River receives freshwater from a set of different tributaries and brooks and runs along intensive agricultural and livestock farm areas and some industrial areas that are dedicated to the textile industry [53]. As can be seen in Figure 1A, the sample sites which present the highest concentration and major distribution of PFASs are the last four points (6, 7, 8 and 9) before the mouth of the river. It is noteworthy that the textile industry has an impact on this area. Moreover, between points 6 and 7, there is one of the major WWTPs along the river. In this area, during the dry season, the sampling sites presented between 3 and 5 different compounds in the dry season, while sites 4 and 5 presented two compounds (PFOS and PFDA) but with higher concentrations in point 5, showing the influence of the WWTP between these two points. In all the samples, PFOA was detected during the dry season but at concentrations between 4 and 18 ng L^−1^. In the wet season, only PFOA was detected and was present in eight samples, but at much lower concentrations (below 5 ng L^−1^).

#### 2.1.2. Leça River

In the 48 km length of the Leça River basin, it has three distinct sections: the upper source and mid-way areas with rural activity (points 1 and 2, respectively); the urban and industrialized area (point 3); and the final section which is impacted by urban discharges and intense industrialization (points 4–9) [54]. The compounds that were more frequently found and at higher concentrations were PFOS, PFOA and PFBS (Figure 1B). Despite the discontinuation in the production of 8-carbon chain compounds, PFOS and PFOA were commonly found, due to their high persistence. PFOS was detected in 6 of the 8 sampling sites in the Leça River during the wet season, but at low concentrations ranging from about 1 to 6 ng L^−1^. On the other hand, they were detected during the dry season in 5 sites, but at slightly higher concentrations. PFOA was present in all the samples during the wet season and in 4 samples during the dry season. PFBS, which is used as a replacement for PFOS, was present in 6 of the 8 sampling sites (i.e., in sites 3 to 8), but only during the wet season and at low concentrations (less than 2 ng L^−1^). 

#### 2.1.3. Antuã River

The Antuã River is a relatively short river with a length of about 30 km. Most of its route can be characterized by an urban and industrial profile, particularly the final section (sites 7–9), which is impacted by footwear, textile, paper production and chemical industries [54]. For this reason, this section presented the highest mean concentration of the total PFASs during the wet season and, in the dry season, was the third most polluted, but at similar levels to those found in the Ave (Figure 1C). PFBS was frequently detected in the wet season in 7 of the 9 sampling sites, but at concentrations below 1 ng L^−1^. PFBAs were present in 33.3% of the samples (sites 1, 2 and 3), but they reached 22.5 ng L^−1^ in site 1. In addition, PFOS and PFOA were the PFASs that were detected at higher frequencies. PFOS was detected in 55% of the samples and PFOA in 100% of the samples in both sampling campaigns. During the wet season, the concentrations were below 10 ng L^−1^, while in the dry season they were slightly higher, up to 17 ng L^−1^. During the dry season, PFNA and PFTeA were quantified in a few samples at low concentrations.

#### 2.1.4. Cértima River

The Cértima has a length of 45 km and in the last 5 km (site 8) becomes a lagoon, flowing into the left bank of the Águeda River. The river crosses intensive agricultural areas and ceramic, metal, and foundry industries [50]. These factors can explain why this river basin was one of the most contaminated by the PFASs included in this study. PFBA and PFOA were the most frequently found compounds (Figure 1D), being present in both cases in 100% of the samples in the wet season and 75% in the dry season. The concentrations of PFBA in both seasons were similar, ranging from about 1 to 9 ng L^−1^. The concentrations of PFOA in the wet season were below 5 ng L^−1^ and in the dry season were below 10 ng L^−1^. PFBS was present in 87% of the samples during the wet season at concentrations below 2 ng L^−1^, while in the dry season it was detected in 3 samples (sites 6, 7 and 8) at concentrations of about 6–8 ng L^−1^. In Cértima River, two of the four predominant compounds, PFBS and PFBA, are replacement compounds for the more persistent ones. 

These facts can be explained by the hydrologic regime of Cértima River, which is of Mediterranean type with dry and hot summers and mild and wet winters, with intense precipitation events. The effect of the summer climate on the hydrologic regime is so severe that, in a typical year, the riverbed dries up for a length of about 20 km, from the source to Mogofores. Then, during the heavy rains, the polar PFASs can be mobilized from surrounding soils to SW. 

### 2.2. Ecological Risk Assessment of PFASs in the Rivers in Portugal 

The adverse ecological impacts of PFASs still require further investigation since these types of compounds are recognized as very persistent and extremely mobile in the environment. Thus, in the present work, the potential environmental risks of PFASs that were detected at higher concentrations, in SW samples, were evaluated based on RQ classification, which is comprised of acute and chronic toxicity (Table 1). The results showed that the RQ values of 8 out of the 9 PFASs were below 0.01, suggesting low risk to organisms at different trophic levels (i.e., fish, daphnid and green algae) in the four target rivers in both seasons. Nevertheless, in the specific case of PFTeA (Figure 2), the RQ values exceeded 1 for acute exposure of fish (96 h) and daphnid (48 h), which indicates a high risk for these organisms. Moreover, a medium risk was suggested for fish and daphnid for chronic exposure and green algae (96 h) for acute exposure, resulting from RQ above 0.1. In fact, the estimated acute aquatic toxicity values (LC50) for this compound are lower than those of the other per- and poly-fluoroalkyl substances targeted in the study, showing that PFTeA is inherently more toxic to aquatic species. Chronic exposure of green algae revealed to be of low risk at the concentrations found. Even though the ecological risk assessment of single substances is valuable and important, the simultaneous exposure to multiple contaminants was not considered. The so-called “cocktail effect” in water matrices is much more complex, i.e., individual compounds that are present at harmless concentrations, in the case of mixtures, may additively or synergistically cause a risk to ecosystems [12,55].

### 2.3. Occurrence of Target PFASs in European SWs

In Table 2, a comparison of the maximum concentrations of PFASs reported in SW from different European studies is presented. As can be seen, the maximum value recorded in this study was in the medium range of most of the studies, except for some that reported punctual high concentrations in France [20], Spain [39], Ireland [56] and Germany [44]. As in the present study, PFOS and PFOA were the compounds that, in general, were more frequently found. Appendix A presents the mean values of an extended survey of the PFASs reported in different rivers in Europe. The values below MLOQ were considered MLOQ/2 and values below MLOD were considered zero. The cluster analysis of these data is presented in the dendrogram in Figure 3. Dissimilarity was calculated using the Bray-Curtis approach and agglomeration using Ward’s method. As can be seen, four significantly different classes can be distinguished. The first class (in purple) presents the centroid levels below 1 ng L^−1^ for most of the compounds, except PFOA. The second class (in green) presents centroids at slightly higher levels and groups the studies of France, Spain, Italy, The Netherlands and some results from Sweden and Austria. The third class (in red) comprises the results from Ireland with the highest centroid levels. Finally, the fourth class (in blue) shows results from basins in Sweden, with the lowest values of the centroids. In summary, the reported values in the present study were in the range of medium-low values of SWs in Europe.

## 3. Materials and Methods

### 3.1. Chemicals and Materials

The 18 reference standards (Appendix A) (i.e., perfluoro-butanoic (PFBA), perfluoro-pentanoic (PFPeA), perfluoro-hexanoic (PFHxA), perfluoro-heptanoic (PFHpA), perfluorooctanoic (PFOA), perfluoro-nonanoic (PFNA), perfluoro-decanoic (PFDA), perfluoro-undecanoic (PFUnDA), perfluoro-dodecanoic (PFDoDA), perfluoro-tridecanoic (PFTrDA), perfluoro-tetradecanoic (PFTeA), perfluoro-hexadecanoic (PFHxDA), perfluoro-octadecanoic (PFODA), perfluoro-butanesulfonic (PFBS), perfluoro-hexasulfonic (PFHxS), perfluoro-octanesulfonic (PFOS), perfluoro-decanesulfonic (PFDS) acids and perfluoro-octane-sulfonamide (PFOSA); ≥98% purity) and the nine surrogate standards (i.e., (^13^C_4_)-perfluoro-butanoic acid (MPFBA (^13^C_4_)), ion (^18^O_2_)-perfluoro-hexane-sulfonate (MPFHxS (^18^O_2_)), (^13^C_2_)-perfluoro-hexanoic acid (MPFHxA (^13^C_2_)), ion (^13^C_4_)-perfluoro-octane-sulfonate (MPFOS (^13^C_4_)), (^13^C_4_)-perfluoro-octanoic acid (MPFOA (^13^C_4_)), (^13^C_5_)-perfluoro-nonanoic acid (MPFNA (^13^C_5_)), (^13^C_2_)-perfluoro-dodecanoic acid (MPFDoA (^13^C_2_)), (^13^C_2_)-perfluoro-decanoic acid (MPFDA (^13^C_2_)) and [^13^C_2_]-perfluoro-undecanoic acid (MPFUdA (^13^C_2_)); ≥98% purity) that were used in this work, were purchased from Wellington Laboratories Inc. (Guelph, ON, Canada). Methanol CHROMASOLV^®^ Plus (HPLC grade), ammonium acetate salt (≥98%) and water used in the instrumental analysis were obtained from Sigma-Aldrich (Steinheim, Germany). For the extraction procedure, methanol (MS grade) was acquired from VWR International (Fontenay-sous-Bois, France), ultrapure water was supplied by a Milli-Q water system and ammonium hydroxide (25%) was obtained from Merck (Darmstadt, Germany). The Oasis^®^ WAX (Weak Anion eXchange) cartridges (3 cc) used for sample preparation were purchased from Waters (Milford, MA, USA). 

### 3.2. Sampling Area, Collection and Preparation

SW samples of four stressed rivers located in Portugal with no crosslinking between them (Ave River, Leça River, Antuã River, and Cértima River; Appendix A) were collected in two different sampling campaigns, namely in the dry (September 2016) and wet (February 2017) seasons. Eight sampling points (SPs) were carefully selected for Leça River and Cértima River and nine SPs for Ave River and Antuã River (Figure 4). These SPs include strategic areas (i.e., near the contamination points that are used for urban, agricultural, or WWTP activities) and close to the source and mouth of each target river. A detailed description of each river can be found in our previous work [50].

All the SW samples were collected by dipping a bottle sampler (properly washed in the laboratory and pre-cleaned with the water from each sampling site) at the centre of the river basin to provide a representative sample. Then, the SW samples were stored at 4 °C in amber glass bottles (1 L) until solid-phase extraction (24 h). SW samples were filtered through 1.2-μm glass-fiber filters (47 mm GF/C, Whatman, Maidstone, UK) before the extraction procedure.

### 3.3. Analysis by LC-MS/MS 

An SPE–LC–ESI-HRMS method was employed for the quantification of the target PFASs, based on a previous method [67,68] with minor modifications. Succinctly, Oasis^®^ WAX cartridges (3 cc) were successively conditioned with 2 mL of methanol and 2 mL of ultrapure water (gravity conditions). Sample loading of 500 mL of SW samples (previously spiked with 10 µL of a standard solution comprising the nine surrogate standards; 2 µg mL^−1^) was carried out at a constant flow rate of 1 mL min^−1^, using a vacuum manifold unit. The cartridges were then dried under vacuum for 15 min. Analytes were eluted in 8 mL (2 mL in four sequent elution steps) of methanol (0.1% NH_4_OH) in PP tubes and evaporated in a Centrivap Concentrator^®^ device (LABCONCO^®^ Corporation, Kansas City, MO, USA), to near dryness. The resulting extracts were transferred from the PP tubes to the LC-vial with 250 µL insert, dried and reconstituted in 100 µL of ultrapure water/methanol (90:10). SPE of blank samples was performed to prevent any cross-contamination during this procedure. All experiments were done in triplicate.

The chromatographic separation was achieved using a Thermo Scientific Aria TLX-1 system (Thermo Fisher Scientific, Franklin, MA, USA) equipped with a C18 analytical column Hypersil GOLD PFP LC (50 × 3 µm) (Thermo Fisher Scientific, San Jose, CA, USA). An extra column, BDS Hypersil C8 (50 × 3) (Thermo Fisher Scientific, San Jose, CA, USA) was used after the LC pumps and before the injection system, in order to delay the contamination from the system pumps. The total run time for each injection (20 μL) was 9 min with a flow rate of 0.4 mL min^−1^. The LC system was coupled to a Thermo Scientific Quantiva triple quadrupole mass spectrometer (Thermo Fisher Scientific, San Jose, CA, USA), equipped with a Heated IonSpray source. All analyses were performed operating in the negative electrospray ionization (ESI(-)) mode and the acquisition relied on selected reaction monitoring mode (SRM) to obtain enough identification points (two transitions for each compound) according to current legislations (Commission_Decision, 2002/657/EC). Selected m/z for each compound can be seen in Appendix A

Linearity, limit of detection (LOD), limit of quantification (LOQ), precision and estimation of measurement uncertainty (MU) were all assessed in accordance with the SANTE/12682/2019 Guidelines [69]. Linearity was evaluated by injecting four calibration points in triplicate into a range from 0.004 to 10 ng L^−1^ for native standards. In accordance with EPA [70], the MLOD was determined using two approaches that were chosen according to the presence or absence of the analytes of interest in the procedural blank. For the first approach, MLODs were estimated as three times the standard deviation of the background concentrations of the procedural blank (*n* = 10). In the second case, the MLODs were determined using the lowest spiking level (*n* = 6). MLOQ was estimated as the lowest concentration of the sample fortified with acceptable precision, by applying the complete analytical method and identification criteria. The MLOQ values are summarized in Appendix A. Precision was obtained by using spiking experiments at three concentration levels (1, 5, 10 ng L^−1^). Each level was analyzed three times on two different days. The inter-day precision was evaluated as the relative standard deviation for each level and the trueness was obtained from the average recovery for each level.

### 3.4. Ecological Risk Assessment

The potential risk caused by the target PFASs to the aquatic ecosystems was estimated based on the ecological risk quotient (RQ) assessment methodology, in accordance with the European Technical Guidance Document on Risk Assessment [71] and the work of various authors [51,55,72], whereby, RQ < 0.1 was considered to indicate low risk; 0.1 < RQ < 1 medium risk; and RQ > 1 high risk for the aquatic environment. The ecological RQ was obtained from the ratio between the measured environmental concentration (MEC) and a predicted no-effect concentration (PNEC), using Equation (1):RQ = MEC/PNEC (1)

The MEC corresponds to the highest concentration found for PFASs in the target rivers in Portugal (refer to Appendix A). PNEC values were determined by using Equation (2), i.e., from the ratio of the acute (LC_50_—Fish (96 h), LC_50_—daphnid (48 h) and EC_50—_green algae (96)) and chronic (ChV) toxicity data, from three different trophic levels, acquired from the ECOSAR (Ecological Structure Activity Relationships) application (version 2.0) and an appropriate assessment factor (AF), considered as 1000 and 100 for acute and chronic toxicity, respectively, in SW [55,71,72,73].
PNEC = (LC_50_, EC_50_ or ChV)/AF (2)

## 4. Conclusions

During two sampling campaigns, which were carried out in the dry and wet seasons, 34 sampling sites, were studied along four river basins in Portugal (Ave, Leça, Antuã and Cértima). PFAS contamination was confirmed in all studied sites in at least one of the studied seasons.

PFOS and PFOA were frequently detected in both sampling campaigns due to their high persistence and extensive use in the past: 40.6 and 41.2% and 82.4 and 97%, for PFOS and PFOA, during the dry and wet seasons, respectively. On the other hand, during the wet season, 74% of the samples presented PFBS and PFBA (another short-chain PFASs) was the compound quantified at the maximum concentration, with 22.6 ng L^−1^ detected in the Antuã River. The rivers presenting the highest levels of contamination by PFASs over the year were Cértima and Antuã, which are impacted by industries such as ceramics, metal-processing, footwear, textile, paper production and chemicals. The mean values of the sum of investigated compounds in all the sampling sites and both campaigns, were 15.9 ng L^−1^ for Cértima and 12.0 ng L^−1^ for Antuã. In addition, in the latter, the mean values of PFASs during the dry season were almost double the values of the wet season due to the dilution effect in the rainy period. However, in the wet season, PFBA and PFBS were the most frequent compounds, and they were detected in sampling sites 1, 2 and 3 but not in these sites during the dry season. Although the Cértima River is tagged as one of the most polluted rivers in Europe due to intense industrial and urban activities, in the present study the Ave River basin presents a mean value of the sum of PFASs of 9.5 ng L^−1^, along the river during both sampling campaigns. However, substantial differences can be highlighted in the dry and wet seasons, with 17 and 1.9 ng L^−1^, respectively. The point source pressures in the Ave are the most dominant threat to water quality in this river basin. 

Finally, comparing the results of selected compounds in the four river basins to those reported by other studies in European SWs, they were found to have similar levels but in the range of medium–low concentrations. Moreover, also in agreement with other studies, high frequencies of detection for PFOS and PFOA were shown, despite their phased-out production and usage.

Concerning the potential for adverse effects on biota, the results of the present study show that the RQ values of eight out of the nine PFASs were below 0.01, indicating low risk to organisms at different trophic levels in the four target rivers in both seasons. Nevertheless, in the specific case of PFTeA, a high risk for fish and daphnids was estimated, as well as a medium risk for green algae (acute exposure). Simultaneous exposure to multiple contaminants was not studied here, but the results of the present study highlight the need to carry out further investigations in this sense in SWs in Portugal.

## Figures and Tables

**Figure 1 molecules-28-01209-f001:**
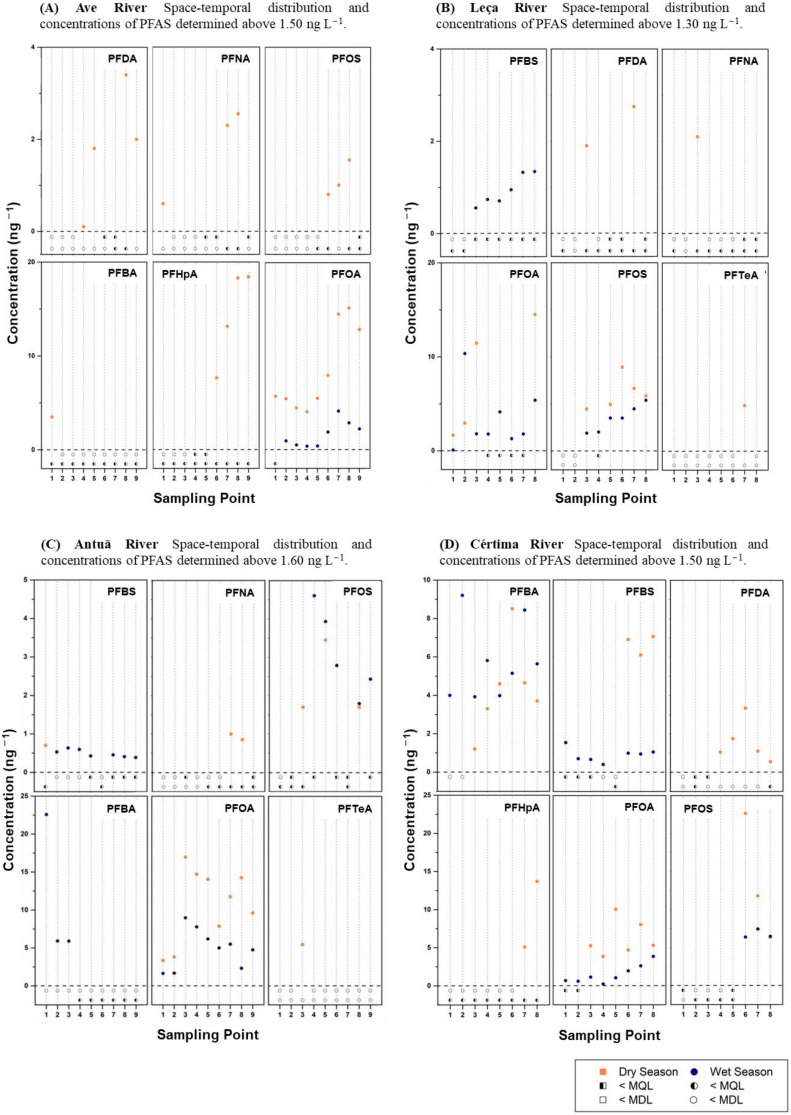
Space-temporal distribution of PFASs and the concentration of compounds found at higher concentrations in the four river basins: (**A**) Ave, (**B**) Leça, (**C**) Antuã, and (**D**) Cértima.

**Figure 2 molecules-28-01209-f002:**
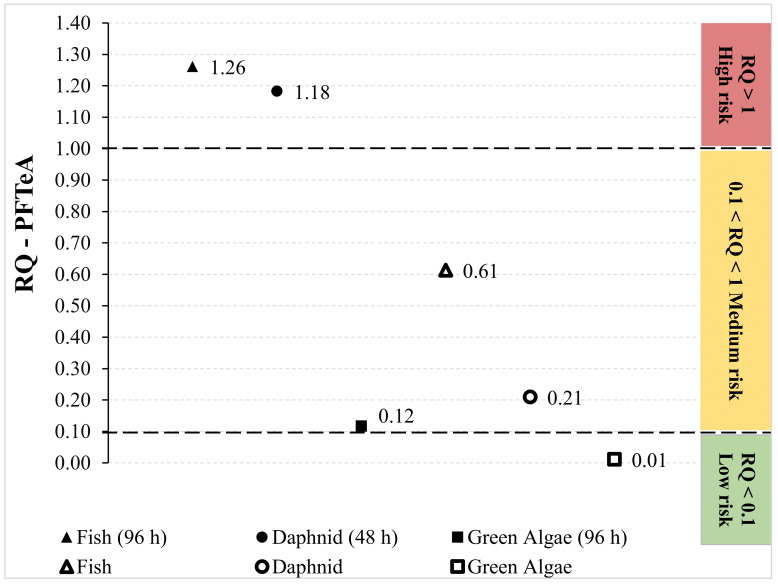
Risk assessment of the higher detected concentration of PFTeA (i.e., Antuã river at SP3 in dry season). The different symbols represent the RQ values for the different organisms and types of toxicity (acute—bold symbols and chronic—open symbols).

**Figure 3 molecules-28-01209-f003:**
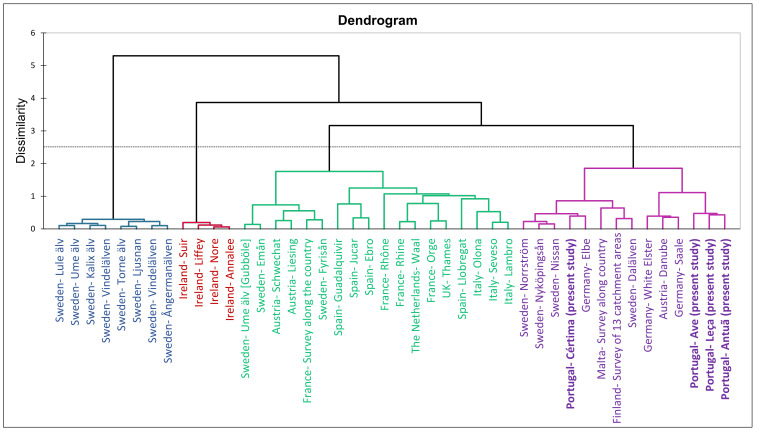
Hierarchical cluster analysis of different studies of PFASs in European SWs. The four river basins of this study are indicated in bold type. References: Sweden [57]; Ireland [56]; Austria [34]; France—Survey along the country [20]; Spain—Guadalquivir and Ebro [58]; Spain—Jucar [59]; France—Rhône and Rhine [46]; The Netherlands [60]; France—Orge [38]; UK—Thames [61]; Spain—Llobregat [62]; Italy [33]; Germany [63]; Malta [64]; Finland—[37].

**Figure 4 molecules-28-01209-f004:**
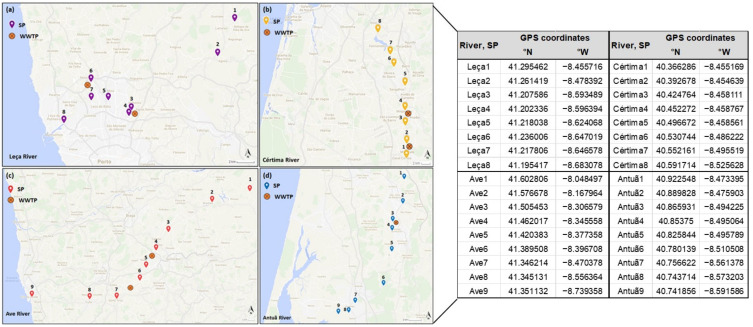
Location of each sampling point (SP) and WWTP in the target rivers: (**a**) Leça; (**b**) Cértima; (**c**) Ave and (**d**) Antuã; and respective GPS coordinates.

**Table 1 molecules-28-01209-t001:** Toxicity data of target PFAS compounds (type of effect, organism, and concentration), predicted no-effect concentration (PNEC), risk quotient and the respective classification obtained.

Compound	Type of Effect	Organism	Conc. (mg L^−1^) ^a^	PNEC (mg L^−1^) ^b^	RQ ^c^	Classification
**PFBA**	Acute (LC50 or EC50 for Green Algae)	Fish (96 h)	1323	1.323	0.000	RQ < 0.1 Low risk
		Daphnid (48 h)	760.6	0.761	0.000	
		Green Algae (96 h)	597.1	0.597	0.000	
	Chronic (ChV)	Fish	131.2	1.312	0.000	
		Daphnid	76.84	0.768	0.000	
		Green Algae	160.9	1.609	0.000	
**PFHxS**	Acute (LC50 or EC50 for Green Algae)	Fish (96 h)	301.3	0.301	0.000	RQ < 0.1 Low risk
		Daphnid (48 h)	190.4	0.190	0.000	
		Green Algae (96 h)	220.4	0.220	0.000	
	Chronic (ChV)	Fish	33.40	0.334	0.000	
		Daphnid	24.98	0.250	0.000	
		Green Algae	73.20	0.732	0.000	
**PFBS**	Acute (LC50 or EC50 for Green Algae)	Fish (96 h)	3597	3.597	0.000	RQ < 0.1 Low risk
		Daphnid (48 h)	2008	2.008	0.000	
		Green Algae (96 h)	1395	1.395	0.000	
	Chronic (ChV)	Fish	344.7	3.447	0.000	
		Daphnid	186.9	1.869	0.000	
		Green Algae	351.9	3.519	0.000	
**PFHpA**	Acute (LC50 or EC50 for Green Algae)	Fish (96 h)	35.43	0.035	0.001	RQ < 0.1 Low risk
		Daphnid (48 h)	24.52	0.025	0.001	
		Green Algae (96 h)	41.43	0.041	0.000	
	Chronic (ChV)	Fish	4.374	0.044	0.000	
		Daphnid	4.150	0.042	0.000	
		Green Algae	16.86	0.169	0.000	
**PFOA**	Acute (LC50 or EC50 for Green Algae)	Fish (96 h)	10.10	0.010	0.002	RQ < 0.1 Low risk
		Daphnid (48 h)	7.437	0.007	0.002	
		Green Algae (96 h)	16.22	0.016	0.001	
	Chronic (ChV)	Fish	1.341	0.013	0.001	
		Daphnid	1.495	0.015	0.001	
		Green Algae	7.576	0.076	0.000	
**PFNA**	Acute (LC50 or EC50 for Green Algae)	Fish (96 h)	2.837	0.003	0.001	RQ < 0.1 Low risk
		Daphnid (48 h)	2.222	0.002	0.001	
		Green Algae (96 h)	6.258	0.006	0.000	
	Chronic (ChV)	Fish	0.405	0.004	0.001	
		Daphnid	0.530	0.005	0.000	
		Green Algae	3.354	0.034	0.000	
**PFDA**	Acute (LC50 or EC50 for Green Algae)	Fish (96 h)	0.788	0.001	0.004	RQ < 0.1 Low risk
		Daphnid (48 h)	0.656	0.001	0.005	
		Green Algae (96 h)	2.386	0.002	0.001	
	Chronic (ChV)	Fish	0.121	0.001	0.003	
		Daphnid	0.186	0.002	0.002	
		Green Algae	1.468	0.015	0.000	
**PFTeA**	Acute (LC50 or EC50 for Green Algae)	Fish (96 h)	0.004	4.320 × 10^−6^	1.261	RQ > 1 High risk
		Daphnid (48 h)	0.005	4.608 × 10^−6^	1.183	
		Green Algae (96 h)	0.047	4.654 × 10^−5^	0.117	0.1 < RQ < 1 Medium risk
	Chronic (ChV)	Fish	0.001	8.884 × 10^−6^	0.613	
		Daphnid	0.003	2.599 × 10^−5^	0.210	
		Green Algae	0.050	4.963 × 10^−4^	0.011	RQ < 0.1 Low risk
**PFOS**	Acute (LC50 or EC50 for Green Algae)	Fish (96 h)	23.66	0.024	0.001	RQ < 0.1 Low risk
		Daphnid (48 h)	16.92	0.017	0.001	
		Green Algae (96 h)	32.65	0.033	0.001	
	Chronic (ChV)	Fish	3.035	0.030	0.001	
		Daphnid	3.131	0.031	0.001	
		Green Algae	14.28	0.143	0.000	

**^a^** data obtained from ECOSAR (Ecological Structure Activity Relationships) application (version 2.0) veloped by U.S. EPA. **^b^** PNEC values were determined by using Equation (2) (PNEC = (LC50, EC50 or ChV)/AF), detailed at “3.4. Ecological risk assessment”. **^c^** RQ < 0.1 indicates low risk; 0.1 < RQ < 1 medium risk; and RQ > 1 high risk for the aquatic environment.

**Table 2 molecules-28-01209-t002:** Comparison of occurrence data for the targeted PFASs in SW samples (ng L^−1^) from this study with previous reports in Europe.

Compound	Max. Concentration Found (ng L^−1^)															
	Present Study	Literature Overview														
	Portugal	Austria [34]	Finland [37]	France [38]	France [20]	Germany [47]	Germany and Netherlands [44]	Germany [39]	Spain [39]	Italy [42]	Malta and Gozo [64]	Netherlands [32]	Norway [65]	Spain [35]	Spain [66]	Switzerland [36]
PFBA	23	<MDL	5.3	<MDL	0.2	3.0	335	0.4	27	n.a.	n.a.	n.a.	8.0	n.a.	0.6	n.a.
PFHxS	1.5	<MDL	6.4	14	217	n.a.	14	<MQL	28	n.a.	2.5	n.a.	1.6	0.8	0.9	14
PFBS	7.1	<MDL	1.5	4.4	29	46	181	<MQL	36	n.a.	n.a.	n.a.	2.0	n.a.	n.a.	7.7
PFHpA	18	<3.2	2.7	4.5	16	11	4.7	24	16	2	11	10	4.4	3.4	10	2.7
PFOA	17	19	5.4	9.4	36	48	41	1.9	35	16	16	>125	5.7	6.3	1.6	7.7
PFNA	2.6	<1.1	23	1.3	30	n.a.	n.a.	<MQL	22	16	3.4	1.5	2.8	0.6	n.a.	<MDL
PFDA	3.4	<1.1	0.5	1.1	10	n.a.	n.a.	<MQL	4.7	11	4.4	3.8	2.0	< 0.82	n.a.	<MDL
PFTeA	5.5	<MDL	<MDL	<MDL	<MDL	n.a.	n.a.	n.a.	n.a.	n.a.	n.a.	n.a.	n.a.	< 0.90	10	n.a.
PFOS	23	35	26	17	370	26	25	0.4	258	39	5.3	n.a.	9.1	5.9	n.a.	60.0

MDL—method detection limit; MQL—method quantification limit; n.a.—not available.

## Data Availability

No new data were created or analyzed in this study. Data sharing is not applicable to this article.

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
