# Peer review of "Per- and Poly-Fluoroalkyl Substances in Portuguese Rivers: Spatial-Temporal Monitoring"

_molecules, 2023, doi:10.3390/molecules28031209_

Round 1

Reviewer 1 Report

This present article reported the per- and poly-fluoroalkyl substances (PFASs) in four river basins in Portugal, and the sampling campaigns were carried out in the dry and wet seasons. The work disclosed the most frequent PFASs in the dry (PFOA and PFOS) and wet (PFOA and PFBS) seasons. The potential environmental risks were also evaluated based on the Risk Quotient (RQ) classification. The research is novel and the article is well organized. Hence I recommend that the paper can be accepted in its current form.

Author Response

Thank you very much for your time and positive response

Reviewer 2 Report

The authors performed a two-seasonal monitoring campaign for eighteen PFASs in the surface waters of four river basins in Portugal. Their concentrations and frequency during dry and wet seasons were detected and their potential ecological risks were calculated based on the RQ classification. The manuscript is well written and displayed in a logical way. I think some revision is needed before it can be considered for publication. Some comments are listed below.

1)    Introduction: PFASs are a mixture. Why these eighteen PFASs were selected to perform the two-seasonal monitoring campaign?

2)    Results and discussion: The quality of figures are poor and should be improved by refereeing to some published papers.

3)    Table 1, the sources of the PNEC value and the classification threshold should be given in the revised manuscript.

4)    Section 2.2: The PFTeA with the RQ values exceeding 1 for fish (96 h) and daphnids (48 h), indicating a high risk for these organisms. Please explain the possible reason to cause the high ecological risk.

5)    Figure 4 It would be helpful to give a whole map showing the locations of Ave, Leca, Antua, and Certima rivers and collection samples. Is there any crosslinking between the rivers?

6)    It would be helpful to add some similar works for comparison, such as Environmental Science & Technology, 2022, 56 14350; Chemosensors, 2022, 10, 502.

Author Response

We thank the Reviewer for the thoughtful comments and constructive recommendations, which were considered as detailed below.

1)    Introduction: PFASs are a mixture. Why these eighteen PFASs were selected to perform the two-seasonal monitoring campaign?

Because during the last decade, these were the most commonly used and studied compounds. So, to compare with other studies in other countries and establish the base level information, the selected compounds were considered the most appropriate ones.

2)    Results and discussion: The quality of figures are poor and should be improved by refereeing to some published papers.

Figures have been changed in the new version of the document.

3)    Table 1, the sources of the PNEC value and the classification threshold should be given in the revised manuscript.

The authors agree with the comment, and this information was included in the revised version of the manuscript. Please see footnotes “b” and “c” of Table 1: “b PNEC values were determined by using Equation 2 (PNEC = (LC50, EC50 or ChV)/AF), detailed at “3.4. Ecological risk assessment” section”; and “c RQ < 0.1 indicates low risk; 0.1 < RQ < 1 medium risk; and RQ > 1 high risk for the aquatic environment”.

4)    Section 2.2: The PFTeA with the RQ values exceeding 1 for fish (96 h) and daphnids (48 h), indicating a high risk for these organisms. Please explain the possible reason to cause the high ecological risk.

The authors thank the Reviewer for the comment. In fact, in the specific case of PFTeA, the RQ values exceeded 1 for acute exposure of fish (96 h) and daphnid (48 h), which indicates a high risk for these organisms according to the ecological RQ assessment methodology used in this study (please see Section 3.4, Lines 320 to 323: “The potential risk caused by the target PFASs to the aquatic ecosystems was estimated based on the ecological risk quotient (RQ) assessment methodology, in accordance with the European Technical Guidance Document on Risk Assessment [62], and the work of various authors [56, 63, 64], (…)”). The high ecological risk obtained may be explained by the lowest values of LC50 for PFTeA in comparison to the other PFASs (please see Table 1). In fact, the estimated acute aquatic toxicity values (LC50) for this compound, given by ECOSAR for fish (96 h) and daphnids (48 h) (i.e., 0.004 mg L-1 and 0.005 mg L-1, respectively), is lower than those of the other per- and poly-fluoroalkyl substances targeted in the study, showing that PFTeA is inherently more toxic to aquatic species. Aquatic organisms are highly affected by the presence of this specific compound in the water environment.

The authors recognize that this point should be clarified. Therefore, a new sentence was added to the revised version of the manuscript. Please see Section “2.2. Ecological risk assessment of PFASs in the rivers in Portugal”, Lines 186-189: “In fact, the estimated acute aquatic toxicity values (LC50) for this compound are lower than those of the other per- and poly-fluoroalkyl substances targeted in the study, showing that PFTeA is inherently more toxic to aquatic species.”.

5)    Figure 4 It would be helpful to give a whole map showing the locations of Ave, Leca, Antua, and Certima rivers and collection samples. Is there any crosslinking between the rivers?

The authors agree with the suggestion and a new Figure was included in the Supplementary Material of the revised manuscript, please see “Figure S1. Target Portuguese rivers (Ave, Leça, Antuã, and Cértima) and location of sampling sites.”; and section 3.2. (Sampling area, collection, and preparation), Lines 258 to 259: “SW samples of four stressed rivers located in Portugal, with no crosslinking between them (Ave River, Leça River, Antuã River, and Cértima River; Figure S1) were collected in two different sampling campaigns, (…)”. The aim of this study was to compare 4 different rivers rising in the Portuguese territory and flowing into the Atlantic Ocean, all with a length of ca. 40 km, except the Ave, whose length is approximately double of Leça. The four rivers studied do not cross at any point along their courses, as shown in Figure S1. Even the rivers with closer locations, the Ave River and the Leça River, do not intersect (please see the zoom in Figure S1).

6)    It would be helpful to add some similar works for comparison, such as Environmental Science & Technology, 2022, 56 14350; Chemosensors, 2022, 10, 502.

New references have been included in the new version of the manuscript.